# General practitioner perspectives on factors that influence implementation of secondary care-initiated treatment in primary care: Exploring implementation beyond the context of a clinical trial

Clair Le Boutillier[1,2]*, Haroon Ahmed[3], Vishal C. Patel[4,5,6], Mark McPhail[4,5], Ben Carter[7], Christopher Ward[8], Vanessa Lawrence[1]

1 Health Service and Population Research, Institute of Psychiatry, Psychology & Neuroscience, King's College London, London, United Kingdom, 2 Division of Methodologies, Florence Nightingale Faculty of Nursing, Midwifery & Palliative Care, King's College London, London, United Kingdom, 3 Division of Population Medicine, Cardiff University, Cardiff, United Kingdom, 4 Institute of Liver Studies, King's College Hospital NHS Foundation Trust, London, United Kingdom, 5 Faculty of Life Sciences and Medicine, School of Immunology and Microbial Sciences, King's College London, London, United Kingdom, 6 The Roger Williams Institute of Hepatology London, Foundation for Liver Research, London, United Kingdom, 7 Department of Biostatistics and Health Informatics, Institute of Psychiatry, Psychology & Neuroscience, King's College London, London, United Kingdom, 8 NIHR Clinical Research Network South London, London, United Kingdom

* clair.le_boutillier@kcl.ac.uk

## Abstract

### Background

The Beta-blockers Or Placebo for Primary Prophylaxis of oesophageal varices (BOPPP) trial is a 3-year phase IV, multi-centre clinical trial of investigational medicinal product (CTIMP) that aims to determine the effectiveness of carvedilol in the prevention of variceal bleeding for small oesophageal varices in patients with cirrhosis. Early engagement of General Practitioners (GPs) in conversations about delivery of a potentially effective secondary care-initiated treatment in primary care provides insights for future implementation. The aim of this study was to understand the implementation of trial findings by exploring i) GP perspectives on factors that influence implementation beyond the context of the trial and ii) how dose titration and ongoing treatment with carvedilol is best delivered in primary care.

### Methods

This qualitative study was embedded within the BOPPP trial and was conducted alongside site opening. GP participants were purposively sampled and recruited from ten Clinical Commissioning Groups in England and three Health Boards across Wales. Semi-structured telephone individual interviews were conducted with GPs (n = 23) working in England and Wales. Data were analysed using reflexive thematic analysis.

**Data Availability Statement:** All relevant data are within the paper and its Supporting Information files.

**Funding:** The study was funded by the National Institute for Health Research (NIHR) Health Technology Assessment Programme (NIHR grant reference: 17/32/04). The funders had no role in study design, data collection and analysis, decision to publish, or preparation of the manuscript.

**Competing interests:** The authors have declared that no competing interests exist.

## Findings

Five overarching themes were identified: *i) primary care is best placed for oversight*, *ii) a shared approach led by secondary care*, *iii) empower the patient to take responsibility*, *iv) the need to go above and beyond* and *v) develop practice guidance*. The focus on prevention, attention to holistic care, and existing and often long-standing relationships with patients provides an impetus for GP oversight. GPs spoke about the value of partnership working with secondary care and of prioritising patient-centred care and involving patients in taking responsibility for their own health. An agreed pathway of care, clear communication, and specific, accessible guidance on how to implement the proposed treatment strategy safely and effectively are important determinants in the success of implementation.

## Conclusions

Our findings for implementing secondary care-initiated treatment in primary care are important to the specifics of the BOPPP trial but can also go some way in informing wider learning for other trials where work is shared across the primary-secondary care interface, and where findings will impact the primary care workload. We propose a systems research perspective for addressing implementation of CTIMP findings at the outset of research. The value of early stakeholder involvement is highlighted, and the need to consider complexity in terms of the interaction between the intervention and the context in which it is implemented is acknowledged.

## Trial registration

ISRCTN10324656.

## Introduction

Liver disease is the third commonest cause of premature death in the United Kingdom and is rising in incidence [1]. Portal hypertension is caused by cirrhosis, and variceal haemorrhage (VH) is a common complication associated with portal hypertension. While the management for varices and variceal haemorrhage has markedly advanced over the past decades, the mortality rate of acute VH is 10%-20% [2]. Prevention of VH is therefore vital in those who have varices. Non-selective beta-blockade (NSBB) has been found to reduce the rate of progression to bleeding or progression to larger varices when initial variceal size is moderate-large varices (>5mm in diameter) [3]. The Beta-blockers Or Placebo for Primary Prophylaxis of oesophageal varices (BOPPP) trial is a 3-year phase IV, multi-centre randomised controlled trial (RCT) that aims to determine if there is a benefit of using carvedilol in the prevention of variceal bleeding for those patients with small oesophageal varices (OVs).

Carvedilol treatment for VH is by endoscopic criteria, assessed following detection of small OVs at gastroscopy. Where required the dose will be up- or down- titrated at clinician discretion, or if the patient reports side effects. Surveillance endoscopy will be undertaken annually. NSBBs are low cost, easy to administer, and do not require specific management expertise. The aim is for dose escalation to be managed in primary care after the trial is completed.

This qualitative study explores General Practitioner (GP) perspectives on factors that influence implementation beyond the context of the trial and investigates how the treatment is best

delivered in primary care. Early involvement of GPs on their views of dose-titration and management of carvedilol provides insights for future implementation, fosters future engagement, and creates an opportunity to enhance the secondary/primary care interface.

## Methods

Ethical approval was obtained from the York and Humber (Leeds West) Research Ethics Committee and the Health Research Authority (REC reference 19/YH/0015).

### Study design

The study used qualitative methodology to provide insights into the complexities of implementation beyond the context of the trial. Semi-structured individual interviews were conducted to gain an understanding of the barriers and facilitators to implementation in primary care. GP participants were recruited from ten Clinical Commissioning Groups (CCGs) in England and three Health Boards across Wales, chosen to provide a mix of different regions and practice size. GPs were selected purposively on the basis that they could offer a particular perspective, that is, as members of the Royal College of General Practitioners, on implementation in primary care. GPs were not recruited on their level of knowledge or experience of managing cirrhosis because the research focused on routine implementation beyond the context of a clinical trial.

Potential GP participants were first approached by the local NIHR Clinical Research Network (CRN) Research Delivery Manager or local contacts, and subsequently recruited by the lead author via telephone or email. Informed verbal consent to enter the study was sought from each participant only after a full explanation and information leaflet was given and time allowed for consideration.

### Data collection

Interviews used open-ended questions to offer the opportunity to share opinions, and to gather a rich understanding of GPs' perspectives. The interview schedule was informed by existing implementation literature and earlier qualitative research that explored patient and secondary-care provider perspectives on barriers and enablers associated with the proposed intervention [4].

Interviews explored GP views on the acceptability of the carvedilol treatment as well as possible barriers and enablers associated with implementation in primary care (i.e., concerns around dose titration, optimal timing of primary care involvement, the role of other primary care professionals (e.g., community pharmacists and practice nurses), and the information, support and infrastructure required to enable success). The interview schedule was revised iteratively in response to the priorities and concerns of participants. The interview schedule is included in Online Data Supplement 1.

Interviews were conducted by telephone and were audio recorded and transcribed verbatim. With participant consent, the audio-files were transcribed by an external transcription company, who are an existing King's College London supplier, and who signed a confidentiality agreement. Interviews were conducted by the lead author between October 2020 and April 2021. GPs were re-imbursed for their time.

### Data analysis

Inductive reflexive thematic analysis was used for data analysis [5,6]. This deliberate and systematic process aims for depth in the interpretation of the data, where codes evolve through a

dual process of thorough engagement and distancing. Researcher subjectivity is an integral component of analysis, and a reflexive diary was kept throughout the study to consider interactions with participants, to detail initial thoughts or assumptions, and to record developing insights [5].

Data analysis began with case-by-case repeated reading of interview transcripts and re-listening of sound files to become immersed in the data. This was followed by line-by-line coding, where individual extracts were grouped under one or several codes. Each code was refined, and where data and interpretation allowed, further codes were developed. Visual maps were used to cluster codes according to connections in the data by considering the patterns between them [7]. An initial coding frame was developed, and analysis was then undertaken to group the codes into overarching themes and sub-themes, that is, shared ideas or concepts. Refinements to the specifics of themes, and thematic patterns continued until a useful and meaningful analysis was achieved. Thematic saturation occurred when the themes had been fully explored and new data was easily accommodated within them [5].

Data analysis occurred using NVivo QSR International qualitative analysis software (version 12). The lead author directed the analysis. Themes were reviewed by two additional raters (HA and VL) to provide an opportunity to reflect on the coding approach, and to enhance the interpretive depth of the data [5].

## Results

### Participants

A total of 23 individual interviews were conducted. GP characteristics are shown in Table 1.

Participants were recruited via nine CRN sites in England and through local contacts in Wales. Of the participants contacted by the researcher (CL), 74% agreed to take part. 26% of those invited to take part did not respond to the invitation. One participant was able to share their previous experience of being asked to prescribe carvedilol for small OVs. The mean duration of the interviews was 19.5 minutes with the time ranging between 12–32.5 minutes.

Five overarching themes were identified. Most participants felt that *primary care is best placed for oversight*. The second theme, *a shared approach led by secondary care*, refers to the recommended pathway to implementation and a desire to feel supported and to deliver the intervention as a part of a wider team. This is complemented by the third theme, *empower the patient to take responsibility*. The fourth theme, *the need to go above and beyond*, represents the potential barriers and challenges to implementation and the final theme, *develop practice guidance*, refers to the specific information and training requirements requested by GPs to enable implementation. Involving GPs in conversations about delivery of a potentially effective treatment at the outset was also identified as a facilitator to implementation:

> *I think it is really good that you are speaking to GPs early, because I think the risk, in terms of ensuring that this is implemented if indeed the trial is successful, I think it is really important . . . if [GPs] are going to be having these dose titrations and discussions, I think it is really important to have them engaged early on. Because otherwise I think it creates a further barrier if you come when the trial finishes and you are trying to then almost push this into primary care, then there may be more apprehension in doing it. So, I think that is a really positive thing.* K64

Due to word limitations, the overarching themes are reported in this paper. The full coding framework (including sub-themes) is available in ODS 2, and a visual illustration of themes and their relationships is included in ODS 3.

**Table 1. Participant characteristics.**

| n (%) | GPs<br>n = 23 | |
|---|---|---|
| **Gender** | | |
| Male | 15 (65.2) | |
| Female | 8 (34.8) | |
| **Age** | | |
| 31–40 | 10 (43.5) | |
| 41–50 | 10 (43.5) | |
| 51–60 | 2 (8.7) | |
| 61–70 | 1 (4.3) | |
| **Time since qualification** | | |
| 0–2 years | 1 (4.3) | |
| 2 years + -5 years | 4 (17.4) | |
| 5 years + -10 years | 6 (26.1) | |
| 10 years + -15 years | 7 (30.5) | |
| 15 years + -20 years | 2 (8.7) | |
| 20 years + -25 years | 2 (8.7) | |
| 25 years + | 1 (4.3) | |
| **Time in current post** | | |
| >6 months | 2 (8.7) | |
| 6–24 months | 3 (13.1) | |
| 2–5 years | 6 (26.1) | |
| 5–10 years | 5 (21.7) | |
| 10–15 years | 4 (17.4) | |
| 15–20 years | 0 (0.0) | |
| 20–25 years | 2 (8.7) | |
| 30 years | 1 (4.3) | |
| **Ethnicity** | | |
| White British | 11 (48.1) | |
| White Other | 2 (8.7) | |
| Asian/Asian British-Indian | 5 (21.7) | |
| Asian/Asian British-Pakistani | 1 (4.3) | |
| Asian/Asian British-Other | 1 (4.3) | |
| Mixed. White & Asian | 1 (4.3) | |
| Mixed. White & Black African | 1 (4.3) | |
| Not otherwise specified | 1 (4.3) | |
| **Current geographical area of work** | | |
| London & South East | 7 (30.5) | |
| East Anglia | 1 (4.3) | |
| East of England | 1 (4.3) | |
| South West | 2 (8.7) | |
| Midlands | 3 (13.1) | |
| North West | 1 (4.3) | |
| Wales | 8 (34.8) | |

## Theme 1: Primary care is best placed for oversight

While the proposed treatment is secondary care-initiated, almost all GPs felt that primary care is best placed to provide oversight, and that the request to manage carvedilol in primary care fitted with the ethos of proactive general practice. This corresponds with GP descriptions of preventative medicine as core work for primary care. The need to reduce risks of untreated varices was acknowledged, with improved quality of life, reduced demand for healthcare further up the system, and cost savings identified as benefits of implementation:

> . . . *if by picking it up means that there's an impact to the patient's life. . . if it means that there's an impact to cost implications for that patient's healthcare as well, so if there are multiple benefits to doing this and managing this in the community early. . . then obviously everyone would be up for that.* K66

GPs went on to explain how they deliver holistic care by working with the whole person within the wider context of their environment. They explained that patient discussions that balance risks and benefits of treatment and take account of other health conditions are a part of day-to-day primary care practice. They reported that discussions around treatment for varices would fit within this scope:

*. . .having those pragmatic discussions, risks versus benefits, taking into account other health conditions. Obviously, GPs, we do that day in, day out, whereas specialists focus on their area. So, in terms of having that holistic all-round view, I think GPs are best placed.* K64

GPs also identified themselves and general practice as a constant in patients' lives. GPs reported that they have existing and often long-standing relationships with their patients, beyond those in secondary care, and these relationships can also be used to support patients in their treatment. GPs went on to explain the structure of the system whereby they can draw on the skills of health care practitioners within their Primary Care Network (PCN) to support oversight: *GPs would probably be the person to oversee this all, but there might be other people. . . if you've got a community pharmacist, they're invaluable as well* (K71). Another explained, how the team can draw on existing systems, processes and expertise of other clinicians within the network to support implementation. For example, pharmacists or nursing staff might monitor blood pressure and check the heart rate of those people taking carvedilol and refer this information to the GP:

*The nurses are very good at monitoring people, because the thing that would have to be monitored would be the blood pressure, so I think they could just treat it in exactly the same way they treat monitoring the blood pressure for a beta-blocker that they're on for another reason. They're aware of what normal blood pressure limits are. So, they can come in and see the nurse and then [the nurses have] got avenues to ask us for opinions, if that blood pressure is out of range.* K72

They went on to explain that, for many patients, the varices treatment would fit into existing review processes. GPs reported that the request to monitor medication in primary care is not new and is recognised as part and parcel of patient management. GPs explained that medication reviews are standard practice with established pathways, and that starting new medication is something that happens all the time:

*People get discharged from hospital all the time on new medications. An ACE inhibitor, for example, they get started on a low dose and we need up titrate them. This is something that we do all the time. So, there's no reason why it can't work.* K57

Alongside, GPs explained that beta blockers are commonly used and usually well-tolerated safe drugs. This familiarity means that GPs can build on their experience and knowledge of how beta blockers work in different situations.

## Theme 2: A shared approach led by secondary care

While most GPs felt that primary care is best placed for oversight, they explained that partnership working, and a shared approach would best support implementation and workload management. GPs outlined a recommended practice pathway to facilitate safe and effective ongoing care, to support GP confidence in managing the clinical request, and to minimise the burden on primary care. Overall, GPs felt that is would be most helpful for secondary care to

initiate treatment after the OVs have been identified. This would also allow patients to ask the specialist questions before their care was transferred to primary care:

> . . . I think definitely if it had been started in secondary care, that would be helpful, if only because the patient had already been primed with some knowledge about what was the plan and why was the plan, and why it was recommended and so on. And also, to give me a steer on starting doses and up-titration from there. K70

Another GP stated that an approach like shared care would be helpful and that it would ideally be a joint decision to prescribe to ensure patient safety:

> There are shared care agreements for prescribing certain drugs whereby we're sent something to sign, and consultant says he's taking responsibility for this part of it, and we take responsibility for other parts. K77

GPs went on to explain that stabilising the dose in secondary care would also provide time for primary-secondary care communication and for organising the medication supply. One GP explained an approach whereby secondary care transferred care after a 14-day dose adjustment period. GPs explained that the handover from secondary care needs to be clear, and communication is required to ensure there is a plan that explicitly states the actions required from general practice, '*So, just really explicitly putting the plan, we've told the patients to do this, and that it's not too much work for the GP*' (K72). GPs also spoke about the need to have alerts set up for monitoring as well as a direct point of contact to access back-up secondary care support and advice because they have the specialist knowledge about starting the medication:

> . . .having the back-up from the specialist that's involved to know that what you're doing is accepted practice and, it's hopefully relatively safe and effective because, when it really boils down to it as a GP, you just want to know that you're doing your best for somebody in a manner that's safe and doing what's expected, shall we say. K56

## Theme 3: Empower the patient to take responsibility

Empowering the patient to take responsibility was identified as a theme that complements the shared approach led by secondary care. GPs spoke about the value of prioritising patient-centred care and involving patients in taking responsibility for their own health. However, GPs acknowledged the time and resources it can take to engage patients in their care and the impact that engagement can have on medication reviews:

> . . .it can be a bit haphazard when you review patients on medication and this particular group of patients are potentially not great at coming in for medication reviews and stuff like that. So that's a challenge. K58

Another GP went on to speak about the need to acknowledge the impact of the often-social needs (e.g., alcohol use, of no fixed abode) of this patient group:

> . . . in this group of patients are they going to turn up for their medication treatment? Are they going to take the medication regularly? Is there going to be risk to the patient if they take it intermittently rather than regularly?. . . So, it's a lot of that context behind monitoring the drug that might be specific to this group of patients which wouldn't be specific to maybe

*patients coming in for their hypertension monitoring or heart disease, for example, it might just have different issues.* K60

To support patient engagement, some GPs identified the need to discuss the pros and cons of treatment options, with the possible use of decision aids, with patients in order to share decision making. These patient conversations provide an understanding as to why the treatment is being offered, and promote involvement in self-managing health. GPs reported this as particularly important when working with people who are asymptomatic and sometimes reluctant to add another drug with possible side-effects:

*. . . discussions about prophylactic treatment that I've had with patients, you're just really trying to work out for the patients themselves what are the benefits. So. . . what's the risk of my varices getting worse or bleeding, versus what are the risks of me taking this treatment.* K60

GPs identified the need for patient education and explained that a patient information leaflet would be helpful to outline the value of the treatment in preventing oesophageal bleeding:

*I think with patients, and especially alcoholics and those suffering from oesophageal varices, it is a pretty horrific thing to happen when they have a bleed. And education to the patient around the importance of taking the betas. . . if I was a sufferer, I wouldn't need much encouragement to take them because I know the consequences of dying in that way. So, some patient education material that quite graphically illustrates to them the benefits of taking it might work.* K57

One GP explained that it would be helpful for the patient to receive a copy of the letter from secondary care to promote a sense of responsibility (and to reduce GP workload) and to advise them to contact their GP for monitoring follow-up:

*. . . quite often we'll get a letter from secondary care saying, "Please [see] the patient" . . . and that's very difficult because it involves us having to get in touch with the patient to tell them, "Please can you come. . .". It's a lot easier if a letter subsequently goes to the patient as well saying, "We've spoken to your GP. Please can you make an appointment to make this happen". That places the burden onto them to get it done.* K57

Some GPs explained that other services involved in the person's care could also be involved in monitoring:

*. . .if you diagnose people with varices and they have a chaotic lifestyle and they don't come and don't engage with GPs then it's a matter of you start the medication and how are you doing to monitor it. . . you do have other community services like we have the GDAS (drug and alcohol service) [and] mental health, they would be better placed to give a bit of advice in addition to all the other things.* K63

## Theme 4: The need to go above and beyond

The most common reported barrier to implementation was the potential that involvement would mean *this is additional work above and beyond what is seen as normal GP work* (K76). Approximately half of all GP participants felt that primary care would be reluctant to take on this request for monitoring because it would create extra work and *I think that one of the*

*barriers is already adding to a particularly busy workload* (K64). GPs explained that while there are existing systems to manage new treatments, it is also important to remain mindful to the capacity of general practice because of *an increasing workload that's been devolved from secondary care* (K77). In relation to this, concerns were also raised about how GPs might find a way to prioritise the treatment strategy:

> *you feel like there's just always 100 plates you're spinning. . . and different drugs and different treatments are fighting to be heard in terms of becoming normal practice. . . I've worked in a gastroenterology department, and I can appreciate how, in the world of gastroenterology if this becomes the new exciting thing then that's quite exciting. But obviously with my GP hat on, gastroenterology is one small part of what feels like hundreds of different things that we're trying to stay on top of.* K56

On the contrary, one GP explained, *it's not going to completely overload GPs because there probably aren't many patients* (K72). This infrequent need to monitor varices was also seen as a barrier to implementation. One GP spoke about the influence of irregularity on implementation and questioned the relevance of the treatment to their practice because varices are not a common problem. They went on to explain, i*n all the time I've worked, I've probably [seen] two patients with oesophageal varices* (K67). For another GP, the irregularity and unfamiliarity prompted concern around the amount of time it would take to monitor the treatment:

> *We don't see these kinds of things that often, so when you do have a patient and you're monitoring them, and following them through, initially it's a bit more intensive, because you're having to get up to speed with whatever the monitoring process is. . . Because it's not like, a big part of day-to-day practice.* K62

Three other GPs went on to state that the unfamiliarity, and their limited experience and knowledge in the area would influence their confidence of managing varices in primary care:

> *And, if I'm honest, because we do it so infrequently, there'd be a part of me that would feel a bit anxious about doing things beyond my capabilities.* K66

One GP felt that due to the specialist nature of the request, management should remain in secondary care:

> *I think the absolute ideal is nurse led service at the hospital. That's ideal. Then the risk is taken away from general practice. . . I think this is a specialist condition and I think . . . there's a major gain to be made that if we can find a medication that destresses. So, I think this is something that you don't want to really want to risk someone not having enough specialist advice on. It's a condition that probably needs specialist management.* K74

A financial incentive was seen as a facilitator for titration to take place in primary care; six GPs spoke about the need for financial reimbursement because they are going above and beyond to support varices treatment in primary care:

> *. . .it's an element of remuneration for that work because the set-up within general practice being private businesses, there's those conflicting elements of clinical as well as, is this going to be seen as part of a GP's normal general medical services contract, or is this above and beyond what we would normally expect GPs to be doing?* K76

On the other hand, one GP stated, *I don't think it would be a huge resource implication, but you will of course have GPs that feel that any additional workload ought to be funded* (K77). Another GP explained how the paperwork for funding needs to be proportionate to the number of patients being seen:

*We have the same thing with coeliac disease, they have these long protocols for relatively few patients, and the claims process and paperwork related to that, it was more laborious than the income. And so, it's the numbers that were important. Although every little bit helps, it's just about being sensible, I think.* K65

## Theme 5: Develop practice guidance

GPs spoke about specific information and training requirements that would support implementation. GPs requested a clear management plan with instructions from secondary care that outlines what to do and how to do it, individualises the dose titration/monitoring needs for each patient and is integrated into the patients notes. Additionally, GPs requested guidance and information on how frequently blood pressure and pulse monitoring is required along with the acceptable thresholds:

*As long as we've got clear guidance as to this blood pressure's acceptable, you can go up to this dose. I think, we do titrate up in other beta blockers in general practice ourselves anyway. But as long as it's clear guidance as to what the schedule is for titrating up it's doable in general practice. . . I don't think it would be that much of a challenge.* K73

Alongside, some GPs requested information on risk reduction:

*So having clear information from the study on what the relative risk reduction and the absolute risk reduction in patients would be vital, not just saying, "Oh this benefits patients". You want to know well by how much? And is that a relevant population to the population that I'm seeing.* K78

One GP suggested that an online link to more information (i.e., the evidence base) or a link to a dedicated website could be included within the secondary care letter. Another explained the need to include the important information on the first page of the letter:

*. . .if you sent a letter with these kind of instructions in them, I think they should be very clear and ideally on the first page because as one of the things we have to do as GPs is look through a lot of letters. And a lot of letters have no information. . . they're updating us and things like that. . . Just because of the workload pressures and just trying to get through the letters as quickly as possible. So, I think if the letter has things to do I think it should be clearly stated early on, in bold, GP action, and then this. And then you can have the detail of the letter after it.* K75

Some GPs felt that the evidence base could be reported and included in NICE and/or British Society for Gastroenterology or Hepatology guidance, BNF, and added to an existing online guideline repository:

*there's something locally that's been developed in the last year, it's called health pathways. . . in that lots of secondary care doctors are writing specific guidelines for primary care, so that it*

*is just quite easy to access, it's online, it's a repository of guidelines that we follow, and it's a great first point of resource for us.* K62

Additionally, some GPs felt that a medication pathway could be developed for commissioning:

*There might be a discussion to be had. . . there's a medicines group for each CCG. It might be worth having a discussion about how they perhaps could safely monitor these in the community.* K71

GPs also highlighted the need to be aware of information patients had received to support ongoing care, and to involve patients and health care professionals in decision aid development:

*If you go down the route of decision aids or anything like that, then because the decision aid is obviously a shared decision, it's two parties involved, then I think having patients involved would be crucial. But also, the healthcare professionals that are going to have that discussion, it would be really useful to get both perspectives.* K64

Some GPs felt that with the right information, training might not be necessary. Others felt that optional training would be helpful, depending on their experience and confidence. Online training was identified as quick, easy, and accessible:

*I think the training for this sort of thing would work best as an e-module with a pre-assessment test [to] measure knowledge as it stands, and then taking you through an evidence-based guideline, illustrated with case scenario-type issues, and then a post-module assessment at the end, just to check my understanding, my learning. I can imagine that working for this kind of thing.* K70

## Discussion

The aim of this study was to explore GP perspectives on factors that influence implementation of secondary care-initiated treatment trial findings into routine primary care. Our inductive approach to analysis found five overarching themes: i) *primary care is best placed for oversight, ii) a shared approach led by secondary care, iii) empower the patient to take responsibility*, iv) *the need to go above and beyond*, and v) *develop practice guidance*. Implementation studies typically classify interventions as events in systems and identify factors that impact uptake across multiple levels of healthcare, including patient, provider, organisation, and policy levels [8,9]. We use the Consolidated Framework of Implementation Research (CFIR) to frame our findings, where four of the five CFIR domains are represented: characteristics of the individual (primary care is best placed for oversight), outer setting (a shared approach led by secondary care and empower the patient to take responsibility), inner setting (the need to go above and beyond), and intervention characteristics (developing practice guidance). Mapping our findings to existing theory informs our developing model for implementing secondary care-initiated treatment in primary care [10].

GPs spoke about their support for implementation and identified that primary care is best placed for oversight. This finding reflects characteristics of the individual and provides an overarching contextual understanding of individual GP professional beliefs and attitudes to

the purpose of their work. This cultural mindset, where proactive general practice is the dominant paradigm, influences individuals' knowledge and belief about the proposed intervention as well as their confidence in delivering the intervention. Almost all GPs identified medicines management as a core part of primary care work and reported a sense of responsibility towards preventing adverse events that will have negative consequences for patients and downstream health services. The core tenets of general practice were described as key to this, including whole-person medicine and continuity of care [11]. While GPs explained that medication reviews are standard practice, a shared approach led by secondary care was identified as the optimal pathway to implementation. This involves partnership working to ensure patients are supported and care is coordinated across the secondary-primary care interface [11]. The CFIR also acknowledges that social capital, that is, staff networks and relationships (within and across organisations) with a collective vision and boundary spanning roles are more likely to support implementation [10]. GPs spoke about the possible development of shared care, whereby they have access to ongoing secondary care specialist knowledge if required, similar to the model used with other chronic conditions [12]. GPs also acknowledged the value and importance of involving patients in this partnership interaction and identified the need for patient information and to prioritise patient-centred care so that patients are empowered and involved in their own care [13]. Consideration to patient needs and resources must be integral to any implementation that seeks to improve patient outcomes [10]. This is especially important given the specific implications of the patient population who are clinically vulnerable because they are living with a chance of VH or death, and for those whose condition relates to alcohol use and/or who have additional vulnerabilities and risks.

Some GPs spoke about potential barriers and challenges to implementation and felt that they would be going above and beyond to support varices treatment in primary care. Barriers to implementation included adding to the primary care workload, the infrequency of the request and subsequent relevance to practice, and the need for financial incentives. These map to two of the inner setting CFIR constructs: implementation climate (compatibility, relative priority) and readiness for implementation. The compatibility of the proposed intervention in terms of how the intervention will impact workload and relative priority of the intervention as in the importance of implementing and relevance to practice, especially given the increase in general practice workload [14]. The need for financial incentives maps to readiness for implementation and access to available resources, including money and training [10]. Characteristics of the intervention were also identified as influencing the success of implementation. Practice guidance was suggested by GPs as a solution (e.g., specific information reporting the evidence base (i.e., evidence strength and quality) and impact on risk reduction) as well as training to guide implementation.

This study promotes research from bench to bedside by engaging stakeholders in implementation at the outset [15]. While the BOPPP trial is a clinical trial of investigational medicinal product (CTIMP) that aims to deliver a 'simple' intervention, there is complexity in implementation, in that the proposed intervention will be managed in and across both secondary and primary care. Using a systems research perspective helps researchers to consider the interaction between an intervention and the context in which it is implemented at the outset of research. The MRC framework for developing and evaluating complex interventions is an established framework that shifts the focus from the binary question of whether an intervention works in the sense of achieving its intended outcome to include questions on acceptability, feasibility and implementation. Early engagement with stakeholders increases the chance of developing an intervention that can be implemented in practice [16].

Involvement of GPs in early conversations about delivering a potentially effective treatment was identified as a support and facilitator for implementation beyond the context of the trial.

GPs spoke positively about the opportunity to offer their perspectives and indicated that participating at the outset of the trial would enhance future uptake and buy-in. GP contributions at this early stage of the research process also provided insights on transferability and implementation that could be considered alongside the trial and as the research progressed. These findings reaffirm the value of engaging intended users (those individuals who will be tasked with implementing the intervention) in the process of exploring factors that influence implementation to promote the success in adopting the intervention [17].

## Strengths and limitations

While the paper extends previous research by exploring patient and recruiting staff perspectives and experiences, it is important to note that the findings are specific to the BOPPP trial. It is possible, however, to enhance transferability by describing the research context and assumptions, and by making connections between the analysis of participants accounts and claims in the extant literature. A strength of the study is the thorough and systematic application of qualitative methods and reflexive thematic analysis [5,6]. While the anonymity of telephone use can allow participants to disclose sensitive information, telephone interviews have received criticism for compromising interviewer/participant rapport and interaction, and for limiting contextual data due to the absence of face-to-face contact and visual cues. However, this method of data collection is convenient, in that it is flexible (in terms of time and location), accessible (i.e., remote research conducted during the COVID-19 pandemic), and allows for a wide reach (e.g., accessing GPs across the UK) [17]. Reflexive thematic analysis and implementation science studies are enhanced by the involvement of trans-disciplinary research teams. Our research team included multi-disciplinary clinicians working in secondary care as well as a general practitioner [6,10].

## Conclusions

The ethos of proactive primary care practice, with preventative medicine as core work provides an impetus for GP oversight. An agreed pathway of care, clear communication, and specific readily accessible guidance on how to implement the proposed treatment strategy safely and effectively are important determinants in the success of implementation. These findings are important to the specifics of the BOPPP trial but can also go some way in informing the wider learning for other trials where findings will impact the primary care workload.

## Supporting information

**S1 Data. GP interview schedule.**
(DOC)

**S2 Data. Full coding framework.**
(DOCX)

**S3 Data. Thematic map.**
(PUB)

## Acknowledgments

We would like to thank all GPs who participated in this study and who generously gave their time and honest thoughts. We are very grateful to the NIHR Clinical Research Network Research Delivery Managers that helped to recruit participants.

## Author Contributions

**Conceptualization:** Clair Le Boutillier, Haroon Ahmed, Vishal C. Patel, Mark McPhail, Ben Carter, Vanessa Lawrence.

**Data curation:** Clair Le Boutillier.

**Formal analysis:** Clair Le Boutillier, Haroon Ahmed, Vanessa Lawrence.

**Funding acquisition:** Haroon Ahmed, Vishal C. Patel, Mark McPhail, Ben Carter, Vanessa Lawrence.

**Investigation:** Clair Le Boutillier, Christopher Ward.

**Methodology:** Clair Le Boutillier.

**Project administration:** Clair Le Boutillier.

**Supervision:** Vanessa Lawrence.

**Writing – original draft:** Clair Le Boutillier, Haroon Ahmed, Vishal C. Patel, Mark McPhail, Ben Carter, Christopher Ward, Vanessa Lawrence.

**Writing – review & editing:** Clair Le Boutillier, Haroon Ahmed, Mark McPhail, Ben Carter, Christopher Ward, Vanessa Lawrence.

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
