## [Decision Letter · Decision Letter 0]

30 Aug 2022

PONE-D-21-39432General Practitioner perspectives on factors that influence implementation of secondary care-initiated treatment in primary care: exploring implementation beyond the context of a clinical trialPLOS ONE

Dear Dr. Le Boutillier,

Thank you for submitting your manuscript to PLOS ONE. After careful consideration, we feel that it has merit but does not fully meet PLOS ONE’s publication criteria as it currently stands. Therefore, we invite you to submit a revised version of the manuscript that addresses the points raised during the review process.

 Please see the comments of two reviewers below. Both reviewers have requested additional clarity in the manuscript. Please ensure that you provide either the COREQ or SRQR checklist or similar when you resubmit, as per PLOS ONE requirements here: http://journals.plos.org/plosone/s/submission-guidelines#loc-qualitative-research, and please ensure that your manuscript presents: 1) defined objectives or research questions; 2) description of the sampling strategy, including rationale for the recruitment method, participant inclusion/exclusion criteria and the number of participants recruited; 3) detailed reporting of the data collection procedures; 4) data analysis procedures described in sufficient detail to enable replication; 5) a discussion of potential sources of bias; and 6) a discussion of limitations.

We look forward to receiving your revised manuscript.

Kind regards,

Hanna Landenmark

Staff Editor

PLOS ONE

Journal Requirements:

“No competing interests”

4**. **Your ethics statement should only appear in the Methods section of your manuscript. If your ethics statement is written in any section besides the Methods, please move it to the Methods section and delete it from any other section. Please ensure that your ethics statement is included in your manuscript, as the ethics statement entered into the online submission form will not be published alongside your manuscript.

Reviewers' comments:

Reviewer's Responses to Questions

**Comments to the Author**

1. Is the manuscript technically sound, and do the data support the conclusions?

Reviewer #1: Yes

Reviewer #2: Yes

2. Has the statistical analysis been performed appropriately and rigorously? 

Reviewer #1: Yes

Reviewer #2: N/A

3. Have the authors made all data underlying the findings in their manuscript fully available?

Reviewer #1: Yes

Reviewer #2: No

4. Is the manuscript presented in an intelligible fashion and written in standard English?

Reviewer #1: Yes

Reviewer #2: Yes

5. Review Comments to the Author

Reviewer #1: This is an analytic study which points to demonstrate the utility of primary and secondary care close involvement in the management of a difficultt to treat patients such as those with a portal hypertension complicated liver cirrhosis.

Few questions to the authors:

- How were patients engaged before being empowered?

- Were the GPs competence on the management of cirrhotic patients certified?

- Somewhere in the text is reported that the nurses should check and monitor blood pressure in patients taking beta-blockers for portal hypertension. To be effective in reducing the portal pressure, beta-blockers should first decrease heart rate by at least 25% from the basic value. Therefore nurses should also check and refer doctors about heart rate. This point needs to be included in the text.

Reviewer #2: This is a well-written manuscript presenting the findings of a qualitative study into GP perspectives of implementing secondary care-initiated treatment in primary care. The results and discussion are clearly presented and place the findings in the context of the wider evidence base. There are a number of areas where clarification or amendment are required:

1. In the abstract, the method of recruitment and sampling approach are not detailed.

2. In some places, carvedilol is written with an uppercase 'C' in the middle of a sentence. Also 'royal college of general practitioners' should be 'Royal College of General Practitioners'

3. In the Methods section, under the heading 'Data collection' the authors should describe how the interview guide was developed and piloted. Did recruitment continue until data saturation was achieved? Also, please add details of who conducted and transcribed the interviews.

4. In the Results, please add the mean duration of the interviews, and the range

5. Did the authors use a reporting guideline for qualitative research i.e. COREQ or SRQR? If so, they should outline where they have reported each of the domains. If not, please consider using one of these reporting guidelines.

6. PLOS authors have the option to publish the peer review history of their article (what does this mean?). If published, this will include your full peer review and any attached files.

Reviewer #1: **Yes: **Ignazio Grattagliano

Reviewer #2: No

---

## [Author Response · Author response to Decision Letter 0]

5 Sep 2022

Dear Editor 05/09/2022

Re: General Practitioner perspectives on factors that influence implementation of secondary care-initiated treatment in primary care: exploring implementation beyond the context of a clinical trial 

Thank you very much for reviewing this manuscript. Please find responses to each point raised by the staff editor and reviewer(s) below. All changes to the manuscript are highlighted with track changes. 

1. Please ensure that you provide either the COREQ or SRQR checklist or similar when you resubmit, as per PLOS ONE requirements.

We have used the COREQ guidelines to report the study – and re-attach the checklist. 

2. Please ensure that your manuscript presents: 1) defined objectives or research questions; 2) description of the sampling strategy, including rationale for the recruitment method, participant inclusion/exclusion criteria and the number of participants recruited; 3) detailed reporting of the data collection procedures; 4) data analysis procedures described in sufficient detail to enable replication; 5) a discussion of potential sources of bias; and 6) a discussion of limitations.

The manuscript presents defined objectives or research questions (p.3), description of the sampling strategy, including rationale for the recruitment method (p.4), participant inclusion/exclusion criteria and the number of participants recruited (p.4 and p.5), detailed reporting of the data collection procedures (p.4), data analysis procedures described in sufficient detail to enable replication (p.5), a discussion of potential sources of bias (N/A to qualitative research); and a discussion of limitations (p.17-18).

We have checked PLOS ONE’s style requirements and have changed the supplementary information file names to meet the requirements. 

4. If you have no competing interests, please state "The authors have declared that no competing interests exist.". This information should be included in your cover letter; we will change the online submission form on your behalf.

Thank you for bringing this to our attention. We can confirm that the authors have declared that no competing interests exist. 

5. We note that you have indicated that data from this study are available upon request. PLOS only allows data to be available upon request if there are legal or ethical restrictions on sharing data publicly. We will update your Data Availability statement on your behalf to reflect the information you provide.

Thank you for clarifying. Our revised Data Availability Statement is: 

This study uses data (containing potentially identifying and/or sensitive information) collected from a small group of staff participants and involves indirect identifiers (such as sex, ethnicity, location, etc.) that may risk the identification of study participants. Sharing data outside of the anonymised excerpts and quotations included in the paper will violate the agreement to which the participants consented. 

We have checked the manuscript and the ethics information has been moved to the Methods section (p. 4).

7. How were patients engaged before being empowered?

Example methods of engagement have been highlighted (p.11): To support patient engagement, some GPs identified the need to discuss the pros and cons of treatment options, with the possible use of decision aids, with patients in order to share decision making. These patient conversations provide an understanding as to why the treatment is being offered and promote involvement in self-managing health. 

8. Were the GPs competence on the management of cirrhotic patients certified?

The GP participants competence on the management of cirrhotic patients was not certified. 

We have clarified this on p.4: GPs were not recruited on their level of knowledge or experience of managing cirrhosis because the research focused on routine implementation beyond the context of a clinical trial.

9. Somewhere in the text is reported that the nurses should check and monitor blood pressure in patients taking beta-blockers for portal hypertension. To be effective in reducing the portal pressure, beta-blockers should first decrease heart rate by at least 25% from the basic value. Therefore nurses should also check and refer doctors about heart rate. This point needs to be included in the text.

This information is contained in a participant quote – as an illustration of how teams might draw on existing systems and processes to support implementation. We have included the above detail to further explain the context of the quote (p.8): Another explained, how the team can draw on existing systems, processes and expertise of other clinicians within the network to support implementation. For example, pharmacists or nursing staff might monitor blood pressure and check the heart rate of those people taking carvedilol and refer this information to the GP.

10. In the abstract, the method of recruitment and sampling approach are not detailed.

The recruitment method and sampling approach have been added to the abstract (p.2): GP participants were purposively sampled and recruited from ten Clinical Commissioning Groups in England and three Health Boards across Wales.

11. In some places, carvedilol is written with an uppercase 'C' in the middle of a sentence. Also 'royal college of general practitioners' should be 'Royal College of General Practitioners'

Thank you for bringing this to our attention. We have corrected using track changes (p.2-4). 

12. In the Methods section, under the heading 'Data collection' the authors should describe how the interview guide was developed and piloted. Did recruitment continue until data saturation was achieved? Also, please add details of who conducted and transcribed the interviews.

We have described how the interview schedule was developed and piloted (p.4-5): The interview schedule was informed by existing implementation literature and earlier qualitative research that explored patient and secondary-care provider perspectives on barriers and enablers associated with the proposed intervention (Le Boutillier et al., 2022) … The interview schedule was revised iteratively in response to the priorities and concerns of participants.

Did recruitment continue until data saturation was achieved? 

We have further explained how thematic saturation was achieved (p.5): Thematic saturation occurred when the themes had been fully explored and new data was easily accommodated within them.

Also, please add details of who conducted and transcribed the interviews.

The interviews were conducted by the lead author between October 2020 and April 2021. This is reported on p.5.

We have added the following detail (p.5): With participant consent, the audio-files were transcribed by an external transcription company, who are an existing King’s College London supplier, and who signed a confidentiality agreement. 

11. In the Results, please add the mean duration of the interviews, and the range.

We have added this information (p.7): The mean duration of the interviews was 19.5 minutes with the time ranging between 12-32.5 minutes. 

Please note we have also updated author affiliations.

We hope that these responses and changes make the paper suitable for publication in PLOS ONE. 

Yours faithfully,

Clair Le Boutillier

---

## [Editor Report · Decision Letter 1]

21 Sep 2022

General Practitioner perspectives on factors that influence implementation of secondary care-initiated treatment in primary care: exploring implementation beyond the context of a clinical trial

PONE-D-21-39432R1

Dear Dr. Le Boutillier,

We’re pleased to inform you that your manuscript has been judged scientifically suitable for publication and will be formally accepted for publication once it meets all outstanding technical requirements.

Kind regards,

Ignazio Grattagliano

Guest Editor

PLOS ONE
---

## [Editor Report · Acceptance letter]

6 Oct 2022

PONE-D-21-39432R1 

General Practitioner perspectives on factors that influence implementation of secondary care-initiated treatment in primary care: exploring implementation beyond the context of a clinical trial 

Dear Dr. Le Boutillier:

I'm pleased to inform you that your manuscript has been deemed suitable for publication in PLOS ONE. Congratulations! Your manuscript is now with our production department. 

Kind regards, 

on behalf of

Dr. Ignazio Grattagliano 

Guest Editor

PLOS ONE